# The basic reproduction number can be accurately estimated within 14 days after societal lockdown: The early stage of the COVID-19 epidemic in Denmark

**Jan Brink Valentin**[1]*, **Henrik Møller**[1,2], **Søren Paaske Johnsen**[1]

**1** Danish Center for Clinical Health Services Research (DACS), Department of Clinical Medicine, Aalborg University and Aalborg University Hospital, Aalborg, Denmark, **2** The Danish Clinical Quality Program and Clinical Registries (RKKP), Aarhus, Denmark

* jvalentin@dcm.aau.dk

## Abstract

### Objective

Early identification of the basic reproduction number (BRN) is imperative to political decision making during an epidemic. In this study, we estimated the BRN 7, 14, 21 and 28 days after societal lockdown of Denmark during the early stage of the COVID-19 epidemic.

### Method

We implemented the SEIR dynamical system for disease spread without vital dynamics. The BRN was modulated using a sigmoid function. Model parameters were estimated on number of admitted patients, number of patients in intensive care and cumulative number of deaths using the simulated annealing Monte Carlo algorithm. Results are presented with 95% prediction intervals (PI).

### Results

We were unable to determine any reliable estimate of the BRN at 7 days following lockdown. The BRN had stabilised at day 14 throughout day 28, with the estimate ranging from 0.95 (95% PI: 0.92–0.98) at day 7 to 0.92 (95% PI: 0.92–0.93) at day 28. We estimated the BRN prior to lockdown to be 3.32 (95% PI: 3.31–3.33). The effect of the lockdown was occurring over a period of a few days centred at March 18th (95% PI: 17th-18th) 2020.

### Conclusion

We believe our model provides a valuable tool for decision makers to reliably estimate the effect of a politically determined lockdown during an epidemic.

**Data Availability Statement:** By Danish law data cannot be shared publicly. Data are available from the Danish Health Data Authority for researchers who meet the criteria for access to confidential

data. Data requests may be sent to the Danish Health Data Authority at forskerservice@sundhedsdata.dk (see also sundhedsdatastyrelsen.dk/da/forskerservice).

**Funding:** The authors received no specific funding for this work.

**Competing interests:** The authors have declared that no competing interests exist.

## Introduction

The SARS-CoV-2 virus has spread rapidly and have already had a dramatic impact on health care systems and societies globally [1, 2]. Moreover, the disease, which is often referred to as the corona virus disease 2019 (COVID-19), has so far caused more than 250.000 deaths world-wide and has had major socio-economic implications in the affected countries [3]. In Denmark the disease has caused more than 500 deaths at the time of drafting, with the first case confirmed on February 27th 2020 [4].

Efforts to reduce or avoid strain on the health care system, as seen in other countries, have been imposed by the Danish government [5]. These efforts have included: home isolation of confirmed cases, closing of schools, non-essential businesses and public workplaces, closing of country borders and restriction of gatherings to no more than 10 individuals. Although case isolation was imposed initially, most of these actions were not presented to the Danish population before March 12[th] 2020 and invoked in the subsequent days, with the final regulations taking effect on March 18th.

So far not much is known about the disease and many of the reported characteristics are based on simplified models [1, 6–10], while other studies focuses mainly on viral load [11] and cell biology including pathogenesis [12, 13]. The amount of disease spread during an epidemic is measured by the basic reproduction number (BRN), however, this number depends on human behaviour and may therefore be different in various cultures and it may change as policy makers impose restrictions on social gatherings [14]. The BRN have previously been estimated in studies on a Chinese population [7, 15], nevertheless, only two studies appears to have investigated the dynamics of the BRN [15, 16]. To our knowledge, no one has yet investigated how early an effect of a political intervention on the BRN can be detected. Hence, the aim of the current study was to determine the time from a political strategy have been enforced until the effect can be accurately estimated. In addition, we aimed to estimate characteristics of the COVID-19 epidemic, such as fraction of infected individuals that are symptomatic and the average infection period.

## Method

### Model

We implemented the so-called SEIR [17, 18] model without vital dynamics. This dynamical system simulates number of susceptible S, exposed E, infectious I and recovered persons R, and is based on the SIR model by Kermack and McKendrick [19, 20], but with an additional equation for modelling the incubation period until cases becomes infectious. In this time period, cases are referred to as exposed, hence, the additional E in SEIR. From number of susceptible, infectious and recovered persons, we calculated daily numbers of hospitalised patients, number of patients admitted to an intensive care unit (ICU) and cumulative number of deaths following viral infection of SARS-CoV-2. These counts were calculated on national level using the parameterisation described below. Model parameters were estimated from individual level patient data when possible, while a few parameters, such as incubation time, were obtained from current literature. We inferred the remaining parameters from number of hospitalised patients, number of patients admitted to an ICU and cumulative number of deaths using a Monte Carlo algorithm.

### Data sources

The Danish National Health Authority provided data on actual numbers of in-patients, ICU patients and cumulative number of deaths in Denmark from March 16[th] 2020 to April 13[th]

2020, both dates included. In Denmark, COVID-19 mortality is reported as infection fatality [4]. Information on age distribution of the Danish population as of January 2020 was obtained from Statistics Denmark, while the North Denmark and Central Denmark regions (1.92 million individuals corresponding to 32.9% of the total Danish population) provided individual level data on their resident patients hospitalised and tested positive with COVID-19 at least once within 14 days prior to and during admission.

Aggregated data is freely available for all in Statbank Denmark. Access to individual patient data can only be obtained by authorized researchers through the Danish National Health Authority, since Danish legislation prohibits unauthorized access.

## Basic reproduction number function

The BRN was modelled over time $t$ using a sigmoid function on the following form:

$$f(t) = a \cdot \text{sigm}(-b(t - k)) + c,$$

where $a+c$ is the BRN prior to intervention, $c$ is the BRN after intervention, $b$ is the transition rate and $k$ is the time at which the intervention is at effect assuming fast transition. The model assumes that the BRN decreases over time, otherwise, $c$ is the BRN prior to intervention and $a+c$ is the BRN after intervention.

## Parameters obtained from individual patient data

Individual level patient data contained information on time of hospital admission, ICU admission, hospital discharge, ICU discharge and death. From these data, we estimated average length of hospital stay and ICU care as well as average time from hospital admission to ICU admission and hospital admission to death.

## Parameters obtained from current literature

We assumed the incubation time to be 5.2 days [21], however, the incubation time is usually defined as time from exposure to symptom onset, while our model relies on the time from exposure to becoming infectious, which is assumed to occur 12 hours prior to symptom onset [2]. Hence, we defined the exposure timeframe to 4.7 days.

The age-stratified proportions of symptomatic cases in need of hospitalisation and intensive care, and age-stratified infection fatality ratios were obtained from the report by Ferguson *et al* [2].

## Parameters obtained by Monte Carlo sampling

We estimated the proportion of cases who are symptomatic, the average time from start of infectious period to hospitalisation, the date of origin of the epidemic, the number of persons initially exposed and the parameters of the BRN function describe above. The date of origin is defined as the date at which the initial person or persons became exposed in Denmark.

The mean generation time, which by definition is equal to the mean infectious period [21], were likewise inferred using the Monte Carlo approach. Finally, we added a parameter describing the ratio of symptomatic cases in need of intensive care compared to those estimated on an external population [2].

## Statistical analysis

Initially, we calculated parameters based on individual level patient data. We estimated average length of hospital stay and ICU care as area under the curve using Kaplan-Meier survival

analysis, while average time from hospital admission to ICU admission and hospital admission to death was estimated, conditioned on patients admitted to the ICU as well as patients who eventually died, respectively. All four parameters were presented with 95% confidence intervals (CI).

Thereafter, we conducted four separate Monte Carlo searches using simulated annealing (SA) with the initial 7, 14, 21 and 28 days of the available data. From here on we will refer to these searches as models 1 through 4. We assumed normally distributed number of in-patient beds, ICU beds and cumulative number of deaths, with a standard deviation of one. The SA factor was varied from an initial value of 200 to a final value of 40. The search was split on 32 threads using a 16-core Xeon(R) CPU E5-2630 v3 @ 2.40GHz hyperthreaded machine. The model parameters with the maximum log-likelihood were chosen as the final model.

Finally, we implemented a second Monte Carlo search for each of the four models using the Metropolis Hastings algorithm. The purpose of this search was to estimate prediction intervals (PI), thus, the algorithm was initiated using the parameters of the best fit, of each of the four models from the prior search. Again, we employed normally distributed number of in-patient beds, ICU beds and cumulative number of deaths, but with a Poisson-like standard deviation. Relevant model parameters as well as projected number of in-patient beds, ICU beds and cumulative number of deaths are presented with 95% PI.

Initial statistical analyses were conducted using Stata 14 (StataCorp. 2015. Stata Statistical Software: Release 14. College Station, TX: StataCorp LP), while we employed Python version 2.7 (Python Software Foundation. Python Language Reference, version 2.7. Available at www.python.org) for the Monte Carlo searches and subsequent analysis.

## Ethics

According to Danish legislation, register studies does not need approval by an ethics committee. Data usage was approved by the local Danish Data Protection Agency.

## Results

From the individual level patient data, we identified 356 patients admitted and tested positive for COVID-19 in the Central Denmark and North Denmark Region with a mean admission time of 10.87 days (95% CI: 9.23–12.51). Of these patients, 80 subjects were admitted to an ICU with a mean length of hospital stay at the ICU of 11.27 days (95% CI: 9.48–13.06), while the mean time from hospital admission to ICU admission was 2.93 days (95% CI: 2.28–3.57). During the observation period, we observed 43 deaths in the Central Denmark and North Denmark Regions, from which we estimated the mean time from hospital admission to death to be 9.02 days (95% CI: 6.76–11.29).

Relevant model parameters are presented in Table 1. All four models found almost the same BRN prior to intervention with the fourth model, which included 28days of observation, estimating the BRN at 3.32 (95% PI: 3.31–3.33). Likewise, models 2 to 4 estimated almost the same BRN after intervention as well as time of intervention, with the fourth model estimating these parameters as 0.92 (95% PI: 0.92–0.93) and March 18[th] (95% PI: March 17[th]-March 18[th]). The first model did not converge and was unable to estimate these two parameters, as the time of intervention could be any time after March 17[th].

The proportion of symptomatic cases and ratio of ICU cases compared to an external population decreased from model 1 to 4, while average number of days from start of infectious period to hospitalisation and average infectious period were stable throughout all four models.

**Table 1. Parameters estimated using simulated annealing.**

|  | Model 1 (7 days) | Model 2 (14 days) | Model 3 (21 days) | Model 4 (28 days) |
|---|---|---|---|---|
| BRN before intervention (95% PI) | 3.37 (3.55–3.86) | 3.35 (3.31–3.39) | 3.45 (3.43–3.47) | 3.32 (3.31–3.33) |
| BRN after intervention (95% PI) | NA | 0.95 (0.92–0.98) | 0.93 (0.91–0.95) | 0.92 (0.92–0.93) |
| Time of intervention (95% PI) | NA | March 18th (17th-18th) | March 18th (17th-18th) | March 18th (17th-18th) |
| Proportion of symptomatic cases (95% PI) | 0.60 (0.49–0.71) | 0.55 (0.47–0.64) | 0.49 (0.46–0.52) | 0.47 (0.45–0.49) |
| Ratio of ICU cases compared to an external population (95% PI)* | 0.95 (0.86–1.05) | 1.02 (0.95–1.10) | 0.91 (0.87–0.96) | 0.85 (0.81–0.89) |
| Average number days from start of infectious period to hospitalisation (95% PI)** | 4.64 (4.08–5.20) | 4.67 (4.43–4.90) | 5.09 (4.75–4.44) | 4.87 (4.39–5.34) |
| Average infectious period (95% PI) | 6.38 (6.22–6.55) | 6.40 (6.33–6.47) | 6.49 (6.47–6.51) | 6.24 (6.15–6.33) |

Columns names indicates data availability, thus, in Model 1 only the initial 7 days of data was used for estimating the parameters, while the entire range of data was used for estimating the parameters in Model 4.

*External population based on the report by Ferguson *et al* [2]

**Hospitalised cases only.

Fig 1 shows the BRN over time estimated by model 4. Assumption of constant levels prior and after intervention is implicit in the parametrisation of the BRN, thus, the PIs are constant at distant times.

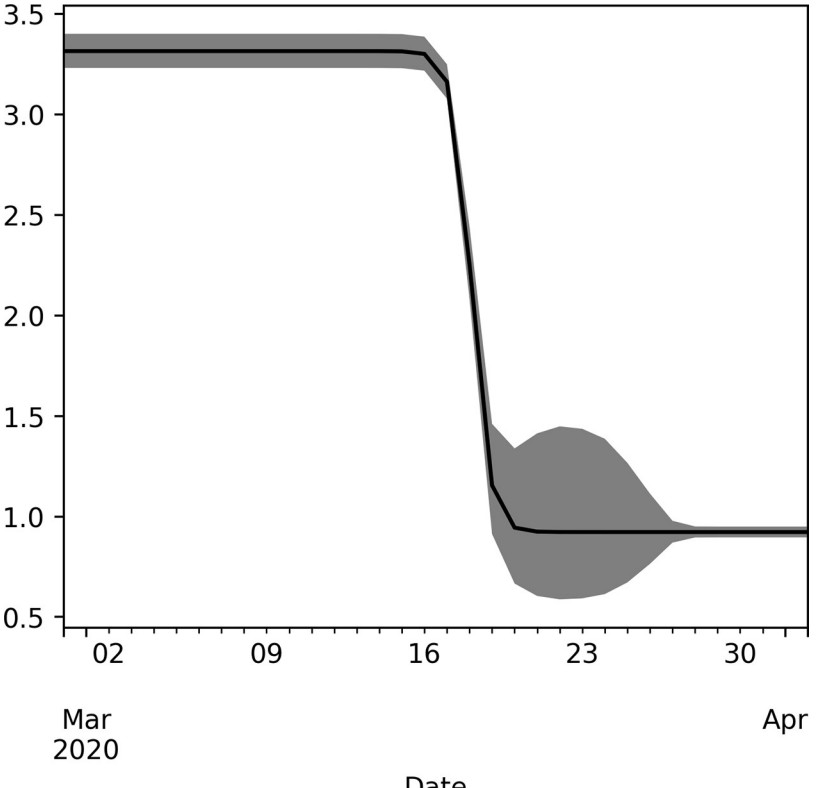

**Fig 1. Estimated change of the BRN with 95% prediction intervals during the early stage of COVID-19 epidemic in Denmark.** BRN is parametrised using a single sigmoid function and prediction intervals are sampled using the Metropolis Hastings algorithm.

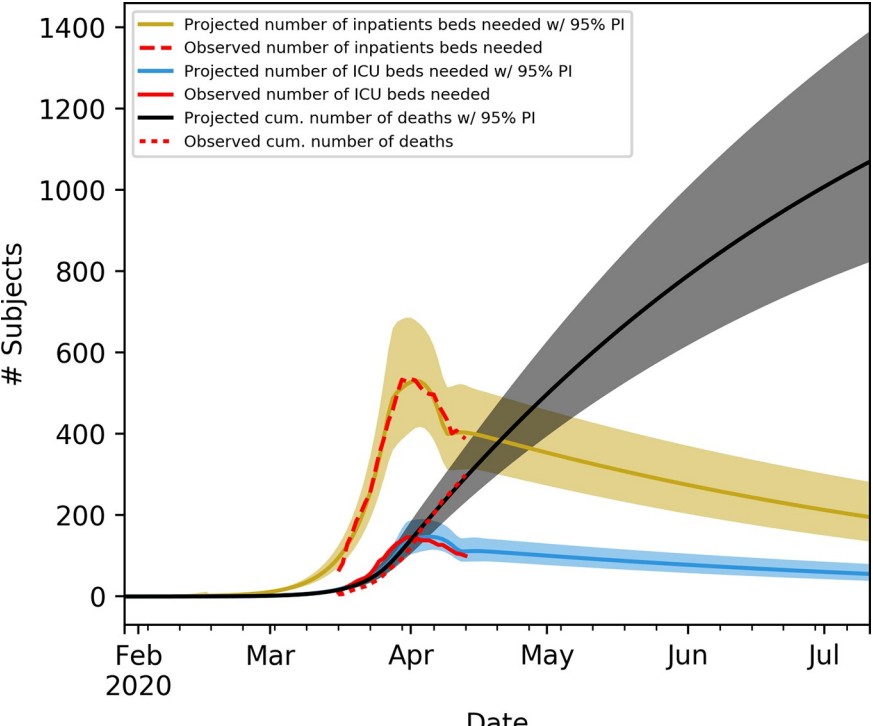

**Fig 2. Observed and projected numbers of in-patient beds needed, ICU beds needed and cumulative number of deaths in Denmark.** Projections follows the SEIR dynamical system with dynamical BRN. Prediction intervals are sampled using the Metropolis Hastings algorithm.

Fig 2 shows projected and observed number of in-patient beds needed, ICU beds needed and cumulative number of deaths in Denmark with 95% PIs. Fig 3 shows projected and observed number of deaths on a daily basis in Denmark with 95% PIs.

## Discussion

To our knowledge this is the first study to infer characteristics of the COVID-19 epidemic based on number of in-patient beds needed, ICU beds needed and cumulative number of deaths, rather than the number of infected persons. In Denmark, health care service is free and available for all residents, and all hospitals are committed to report these numbers to the national health authority. Hence, we consider our outcome measures to be highly reliable, in contrast to number of infected, which is highly dependent on test strategy and consequently also sensitive to changes in test strategy during the epidemic.

Visual inspection of Figs 2 and 3 shows a good fit of the data, though, the number of ICU beds needed seems to be somewhat shifted in time. However, the model assumes that the average time from start of infectious period to hospitalisation is independent of disease severity. An explanation of this bias may simply be that subjects in need of intensive care are hospitalised earlier compared to hospitalised patients in general. This discrepancy may also explain the decrease in proportion of symptomatic cases and ratio of ICU cases compared to external population as more data becomes available.

Our findings suggest that the severity of the epidemic may be higher compared to other studies, as we estimate the BRN to be above 3.3 even after inducing case isolation. A similar study by Kucharski *et al* [15] estimated the dynamics of the BRN in Wuhan using sequential

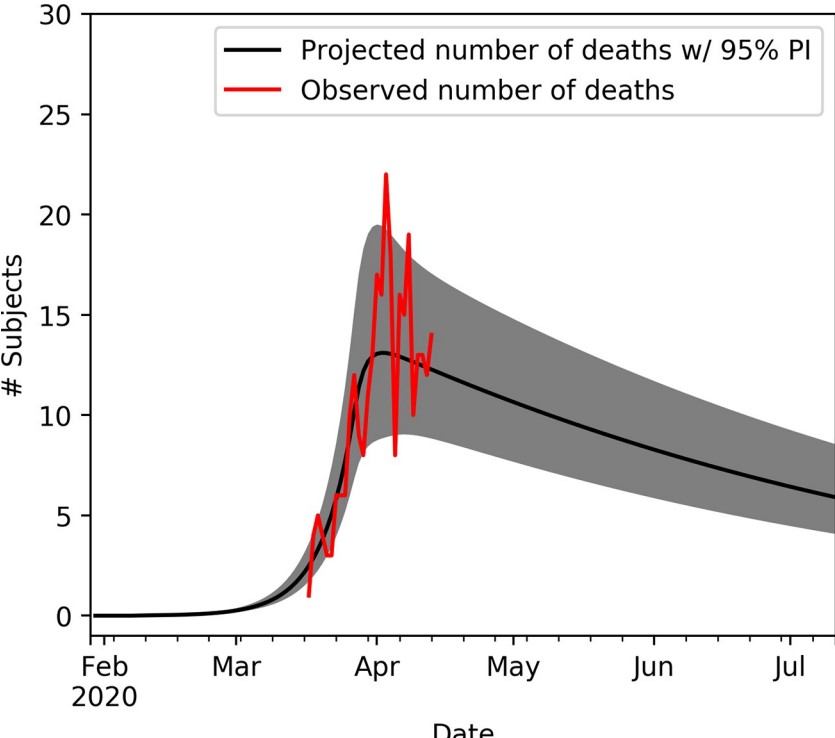

**Fig 3. Observed and projected numbers of deaths in Denmark.** See Fig 2 for further specification.

Monte Carlo simulation [22] and found a BRN of 2.6 at its highest and just below 1.0 at its lowest. However, this model was fitted on number of confirmed cases, which was reflected in the high inaccuracy of the model. Other studies reported a BRN prior to lockdown much larger than the BRN of our study of 5.60 (95%CI: 4.77–7.05) [16] and 6.94 (95%CI: 6.52–7.39) [23]. Both of these studies inferred their parameters on number of COVID-19 related deaths, where the latter also included in-patient counts as well as a latent model for the development in daily new cases. However, none of these models included an incubation period, which may explain the larger initial BRN.

From the individual level patient data, we were able to estimate clinical characteristics based on data from two regions of Denmark. Together with the Monte Carlo estimated parameters we found that the mean time from start of infectious period to hospitalisation was 4.9 days, and with a mean time from hospitalisation to death of 9.0 days, we arrive at infectious onset to death of 13.9 days. Similar to our study, a study by Verity *et al* [1] estimated the time to death conditioned on whether death occurred, but on a population of Hubei, mainland China. They found the mean time from symptom onset to death to be 18.8 days. Although these figures are not fully comparable, the difference is important. The Verity study, however, was based on only 24 deaths. Another study found that the mean time from illness onset to death was 15.0 days and from hospital admission to death was 8.8 days [6].

A study by To *et al* found the infectious time to be 7.5 days (95% CI: 5.3 to 19) [7] with a CI well containing the infectious time estimate of our study, which was 6.24 (95% PI: 6.15–6.33). The same study found the mean duration of symptom onset to hospitalisation to be 9.1 and 12.5 days in two time-stratified populations. In contrast, our study suggests this number to be less than 4.87 days. The difference may be explained by cultural differences between Denmark and China, however, another study in China estimated the median time from symptom onset

to hospitalization to be 1.2 days (range: 0.2 to 29.9 days) [9], while a study from Singapore found this number to be 4 days [8]. Finally, Guan *et al* found the mean duration of hospitalisation to be 12.8 days [10], which is similar to the 10.9 days in our study. There is still a high disagreement between the clinical characteristics of COVID-19 in different studies. Nevertheless, we believe our findings to be generalizable to external populations, because our study is conducted using reliable data in a country where health care service is free and available for all residents.

The model suggests that the change from initial BRN to the reduced BRN is centred around March 18th 2020, and from Fig 1 it seems the change occurred over a period of four days or less. The timing of the change occurred as final government orders were at effect only six days after the societal lockdown strategy was initially presented. Social distancing was encouraged early on, however, fines were not imposed until March 18th. Nevertheless, from the current study is not possible to distinguish which of the political actions were most effective, as all actions were implemented within a short time.

A few of the model parameters is obtained from the literature, which may have some impact on estimated model parameters, considering these may not be fully generalizable to a Danish population. In addition, the model assumes that the time from symptom onset to hospitalisation is the same for severe cases as for in-patients in general. This lack of differential initial symptom load may explain the decreasing fraction of symptomatic cases and ratio of ICU cases compared to the external population, as the model may be unable to accurately predict the number of ICU beds needed. A major strength of the study is the use of in-patient data and death counts rather than only inferring on infection counts of the general population, since we are more likely to capture all severe cases and are not reliant on test strategy. However, in case of differential behavioural patterns following political intervention between the physically vulnerable part of the population and the remaining general population, the results of the current model may be biased. This issue may be solved by implementing a dynamical model for the proportion of symptomatic cases, though the parametrisation of such a model is not straight forward. Moreover, our current model is subject to bias from changes in treatment availability, such as increased use of remdesivir, though current treatment opportunities have limited effect [12, 24].

The current global challenge of reopening society with fewest fatal consequences and at the same time reducing economical costs warrants valid and precise prediction models. We believe that the current study provides a valuable tool for early measurement of the effect of a political intervention. As time progresses and the political strategy is adjusted, one can simply add additional sigmoid function, without having to refit the parameters of the prior BRN functions. However, the results of the current study suggests that each adjustment of any political strategy aimed at constraining the COVID-19 epidemic should be invoked with at least fourteen days intervals.

## Conclusion

In this study we estimated characteristics of the COVID-19 epidemic in Denmark based on the number of in-patient beds needed, the number of ICU beds needed and the cumulative number of deaths. We found the model parameters to be stable as more data were accrued over time. Moreover, we found that the time of change from initial BRN to the reduced BRN to be in good accordance with actual timing of the political actions. We believe the model constitutes a useful tool for early assessment of the effect following political interference. The model can easily be implemented in other settings.

## Author Contributions

**Conceptualization:** Jan Brink Valentin, Henrik Møller, Søren Paaske Johnsen.

**Data curation:** Jan Brink Valentin.

**Formal analysis:** Jan Brink Valentin.

**Funding acquisition:** Søren Paaske Johnsen.

**Investigation:** Jan Brink Valentin.

**Methodology:** Jan Brink Valentin, Henrik Møller, Søren Paaske Johnsen.

**Project administration:** Jan Brink Valentin.

**Resources:** Jan Brink Valentin.

**Software:** Jan Brink Valentin.

**Supervision:** Søren Paaske Johnsen.

**Validation:** Jan Brink Valentin.

**Visualization:** Jan Brink Valentin.

**Writing – original draft:** Jan Brink Valentin.

**Writing – review & editing:** Jan Brink Valentin, Henrik Møller, Søren Paaske Johnsen.

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
