## [Decision Letter · Decision Letter 0]

27 Aug 2020

PONE-D-20-17348

The basic reproduction number can be accurately estimated within 14 days after societal lockdown: The early stage of the COVID-19 epidemic in Denmark

PLOS ONE

Dear Dr. Valentin,

Thank you for submitting your manuscript to PLOS ONE. After careful consideration, we feel that it has merit but does not fully meet PLOS ONE’s publication criteria as it currently stands. Therefore, we invite you to submit a revised version of the manuscript that addresses the points raised during the review process.

We look forward to receiving your revised manuscript.

Kind regards,

Chiara Lazzeri

Academic Editor

PLOS ONE

Journal Requirements:

Reviewers' comments:

Reviewer's Responses to Questions

**Comments to the Author**

1. Is the manuscript technically sound, and do the data support the conclusions?

Reviewer #1: Yes

Reviewer #2: Partly

2. Has the statistical analysis been performed appropriately and rigorously? 

Reviewer #1: Yes

Reviewer #2: I Don't Know

3. Have the authors made all data underlying the findings in their manuscript fully available?

Reviewer #1: No

Reviewer #2: Yes

4. Is the manuscript presented in an intelligible fashion and written in standard English?

Reviewer #1: Yes

Reviewer #2: No

5. Review Comments to the Author

Reviewer #1: the authors' aim was to determine "the time from a political strategy have been enforced until the effect can be accurately estimated". Furthermore they aimed to estimate the fraction of infected individuals that are symptomatic and the average infection period. the primary analysis focuses on basic reproduction number before and after lockdown. The population in study is represented by the Danish National Health authority in-patient registry.

comments:

#1 in the methods section it should be stated if Danish National Health authority in-patient registry data are available on a public repository or not.

# 2 the authors should better address in the discussion a major limitation: the analysis was made on an in-patient population and not int the whole population: how could this bias the results?

# 3 there is a couple of papers that should be interesting to discuss:

- Alexandre Hyafil , David Moriña. Analysis of the impact of lockdown on the reproduction number of the SARS-Cov-2 in Spain. Gac Sanit . 2020 May 23;S0213-9111(20)30098-4. doi: 0.1016/j.gaceta.2020.05.003. Online ahead of print.

- Greg Dropkin. COVID-19 UK Lockdown Forecasts and R 0. Front Public Health. 2020 May 29;8:256. doi: 10.3389/fpubh.2020.00256. eCollection 2020.

Reviewer #2: Thank you for sharing your work. I find the overall theme of your work interesting. However, there are some points that should be addressed. The manuscript needs some language editing. I think with the help of a native speaker, you can signficantly improve the quality of your work. You have chosen to use the number of admitted patients, ICU admissions and deaths to calculate the basic reproduction number for COVID-19 spread; Were all of these patients confirmed as COVID-19 positive? In literature, we read about incubation times of up to 14 days. Yet you chose an icubation time of 4.7 days. Why?

6. PLOS authors have the option to publish the peer review history of their article (what does this mean?). If published, this will include your full peer review and any attached files.

Reviewer #1: No

Reviewer #2: No

---

## [Author Response · Author response to Decision Letter 0]

25 Jan 2021

Editor comments

Response: We have added the following sentence to the Data sources section:

“Aggregated data is freely available for all in Statbank Denmark. Access to individual patient data can only be obtained by authorized researchers through the Danish National Health Authority, since Danish legislation prohibits unauthorized access.”

Reviewer comments

Reviewer #1:

The authors' aim was to determine "the time from a political strategy have been enforced until the effect can be accurately estimated". Furthermore they aimed to estimate the fraction of infected individuals that are symptomatic and the average infection period. the primary analysis focuses on basic reproduction number before and after lockdown. The population in study is represented by the Danish National Health authority in-patient registry.

Comment #1: in the methods section it should be stated if Danish National Health authority in-patient registry data are available on a public repository or not.

Response: Thank you for taking the time to review our manuscript. See response in editor comments.

Comment # 2: the authors should better address in the discussion a major limitation: the analysis was made on an in-patient population and not int the whole population: how could this bias the results?

Response: Thank you for addressing the lack of thoroughly discussing the limitations of our study. We have expanded the paragraph in the discussion section addressing the pros and cons of using in-patient data and deaths counts rather than the infection counts of the general public.

Comment # 3: there is a couple of papers that should be interesting to discuss:

- Alexandre Hyafil , David Moriña. Analysis of the impact of lockdown on the reproduction number of the SARS-Cov-2 in Spain. Gac Sanit . 2020 May 23;S0213-9111(20)30098-4. doi: 0.1016/j.gaceta.2020.05.003. Online ahead of print.

- Greg Dropkin. COVID-19 UK Lockdown Forecasts and R 0. Front Public Health. 2020 May 29;8:256. doi: 10.3389/fpubh.2020.00256. eCollection 2020.

Response: Thank you for expanding our reference list. We have added a paragraph to the discussion section, which addresses results of the suggested publications.

Reviewer #2:

Thank you for sharing your work. I find the overall theme of your work interesting. However, there are some points that should be addressed. 

Comment #1: The manuscript needs some language editing. I think with the help of a native speaker, you can signficantly improve the quality of your work.

Response: Thank you for taking the time to comment on our manuscript. The manuscript has already been language edited by a native English speaker and have now been through a second round of language editing.

Comment #2: You have chosen to use the number of admitted patients, ICU admissions and deaths to calculate the basic reproduction number for COVID-19 spread; Were all of these patients confirmed as COVID-19 positive?

Response: Thank you for your concern. Patients were defined as COVID-19 positive according to the Danish national guidelines, which includes test for active virus (at the time all tests were administered using polymerase chain reaction (PCR) based tests) within 14 days prior to admission or several times during admission or both. Thus, patients may only have been tested positive once. We added a comment regarding this issue to the data sources section.

Comment #3: In literature, we read about incubation times of up to 14 days. Yet you chose an icubation time of 4.7 days. Why?

Response: Thank you for your concern. We agree that the reported incubation time is varying substantially between publications. However, it is not possible to infer the incubation time from our dataset, thus, we reviewed the publications, for which the authors claim to have estimated the incubation time. Based on this review we chose the incubation time from the study, which we believe is the most scientifically rigorous.

---

## [Editor Report · Decision Letter 1]

1 Feb 2021

The basic reproduction number can be accurately estimated within 14 days after societal lockdown: The early stage of the COVID-19 epidemic in Denmark

PONE-D-20-17348R1

Dear Dr. Valentin,

We’re pleased to inform you that your manuscript has been judged scientifically suitable for publication and will be formally accepted for publication once it meets all outstanding technical requirements.

Kind regards,

Chiara Lazzeri

Academic Editor

PLOS ONE
---

## [Editor Report · Acceptance letter]

5 Feb 2021

PONE-D-20-17348R1 

The basic reproduction number can be accurately estimated within 14 days after societal lockdown: The early stage of the COVID-19 epidemic in Denmark 

Dear Dr. Valentin:

I'm pleased to inform you that your manuscript has been deemed suitable for publication in PLOS ONE. Congratulations! Your manuscript is now with our production department. 

Kind regards, 

on behalf of

Dr. Chiara Lazzeri 

Academic Editor

PLOS ONE